# Hepatoprotective effects of Cassiae Semen on mice with non-alcoholic fatty liver disease based on gut microbiota

Hanyan Luo[1,2], Hongwei Wu[1,2], Lixia Wang[1], Shuiming Xiao[1], Yaqi Lu[1], Cong Liu[1], Xiankuo Yu[1], Xiao Zhang[1], Zhuju Wang [1,3 ✉] & Liying Tang [1,3 ✉]

Cassiae Semen (CS), the seeds of Cassia obtusifolia L. and C. tora L, have a long medicinal history in China, with suggestions for it to relieve constipation and exert hepatoprotective effects. However, the underlying mechanisms are still unclear. In this study, mice with high-fat diet (HFD)-induced non-alcoholic fatty liver disease (NAFLD) were used to study the hepatoprotective effects of CS. The relationship between gut microbiota and hepatoprotective effect mechanisms mediated by CS extracts, the total aglycone extracts of CS, rubro-fusarin-6-β-gentiobioside, and aurantio-obtusin were examined. Our data indicate that CS extracts and components confer a protective effect by ameliorating lipid accumulation, intestinal barrier damage, liver damage, and inflammation on HFD-induced liver injury. Meanwhile, fecal microbe transplantation exerted the pharmacological effect of CS on HFD-fed mice; however, the efficacy of CS was inhibited or eliminated by antibiotic-induced dysbiosis. In conclusion, the therapeutic effects of CS on NAFLD were closely related to the gut microbiota, suggesting a role for TCM in treating disease.

[1] Institute of Chinese Materia Medica, China Academy of Chinese Medical Science, Beijing 100700, China. [2] These authors contributed equally: Hanyan Luo, Hongwei Wu. [3] These authors jointly supervised this work: Liying Tang, Zhuju Wang. ✉email: wangzhuju@sina.com; bjtangliying@163.com

Consumption of a high-fat diet (HFD) can cause many disorders associated with fat accumulation, increasing the risk of cardiovascular and metabolic diseases[1]. According to the "second strike" and lipotoxicity theories[2], accumulation of lipid in the liver, resulting in lipid peroxidation, inflammation, oxidative stress, and other hepatocellular damage/death, affecting the normal function of liver. However, as well as fat accumulation and inflammation, it is often accompanied by intestinal damage and imbalanced gut microbiota[3]. By investigating the gut–liver axis, it was observed that gut microbiota changes played important roles in inducing and promoting liver function damage[4,5]. Intestinal microbiota disorders, intestinal mucosal barrier injury, and enterogenous endotoxemia disrupts gut and liver balance, thereby affecting several liver diseases via gut-liver axis[1]. Therefore, reducing lipid accumulation and regulating gut microbiota imbalance are important in maintaining liver health.

Cassiae Semen (CS) is the dried and mature seed of *Cassia obtusifolia* L. and *Cassia tora* L, which has a medicinal history of more than 2000 years in China. As a medicinal-edible material, it is widely used in Asia, especially China, Korea, Japan, and India, to relieve constipation, improve hyperlipidemia, hepatoprotection, and prevent myopia[6–9]. Modern studies have shown that CS elicits good hepatoprotective effects which reduce fat accumulation in non-alcoholic fatty liver disease (NAFLD) mice, improves lipid peroxidation, and alleviates liver inflammation[7,10,11]. Moreover, CS act on the intestine, its laxative effect and mechanism are relatively clear[12–14], but its mechanism of liver protection is unclear. At the same time, based on the preliminary research found, if antibiotics disturbed the complete structure of gut microbiota in the mouse, its liver protecting effect is significantly reduced. Therefore, it was hypothesized that gut microbiota plays a major role in the efficacy of CS. While acting on the intestine, the balance of the gut microbiota is also adjusted, and CS functions via the gut–liver axis to generate hepatoprotective effects. Based on this hypothesis, three parts experiments were designed to verify this. The first study was the pharmacological assessment of CS in HFD-fed mice, to clarify its therapeutic effects on NAFLD. The second component investigated fecal microbe transplantation (FMT) to observe whether FMT could transfer the pharmacological effect of CS to HFD-fed mice. Finally, antibiotic-induced dysbiosis studies were conducted to observe the influence of gut microbiota disorder on CS efficacy. These latter studies illustrated that CS's hepatoprotective effects were related to gut microbiota from two perspectives. The experimental setup is shown in Fig. 1.

The main chemical constituents of CS are anthraquinone, naphthalene, and naphthalopyranone, which exist in the form of glycosides, and these are the main active ingredients for hepatoprotective, hypolipidemic, and antimicrobial activities[15–17]. Glycosides usually need to be digested into aglycones by gut microbiota to be absorbed. Our previous study showed that in 40 °C water, most of these glycosides were enzymatically hydrolyzed to produce aglycones[18]. In order to determine whether aglycones can play a better role than CS extracts, the total aglycones (TA) obtained from CS after enzymatic hydrolysis was administered directly to animals. Two bioactive compounds, rubrofusarin-6-$\beta$-gentiobioside (RG) and aurantio-obtusin (AO), typical naphthalene pyranone and anthraquinones representative components of CS were also chosen.

## Results

### Hepatoprotective effects of Cassiae Semen on NAFLD mice

*Cassiae Semen improves fat accumulation in the liver of NAFLD mice.* During NAFLD pathogenesis, lipid accumulation is an important indicator of liver injury[2]. To determine the effects of CS and several of its bioactive compounds on hepatic fat accumulation,

C57BL/6 mice were fed a HFD for up to 17 weeks (119 days), and then treated with Cassiae Semen extract (CSE) (10 g kg⁻¹), the total aglycone extract of CS (TA) (10 g kg⁻¹), rubrofusarin-6-$\beta$-gentiobioside (RG) (20 mg kg⁻¹) and aurantio-obtusin (AO) (20 mg kg⁻¹) from week 14 to the end of week 17 (21 days).

Data indicated that the liver weight index, total cholesterol (TC), triglycerides (TG), free fatty acids (FFA), and low-density lipoprotein cholesterol (LDL-C) levels of HFD-fed mice decreased after CS extracts or RG and AO administration (Fig. 2a–e). While in all treatment groups, there was no significant difference in high density lipoprotein cholesterol (HDL-C) levels than HFD group ($P > 0.05$). Oil red staining revealed reduced lipid accumulation in the liver of these mice after comparison with the HFD group (Fig. 2g). Moreover, data indicated that CSE and TA exerted improved effects (better than two compounds, RG and AO) on fat accumulation (TG and FFA) in NAFLD mice, while there was no significant difference between CSE and TA, except for TC.

*Cassiae Semen ameliorates liver injury and inflammation in NAFLD mice.* Long-term HFD mice experienced some liver damage, but after CS extracts or compounds were administered, serum alanine aminotransferase (ALT) and aspartate aminotransferase (AST) levels decreased (RG and AO have little effect on AST) (Fig. 2h, i). Increased tumor necrosis factor-alpha (TNF-α) and interleukin (IL)−6 levels in HFD-fed mice liver homogenates suggested that a long-term HFD inflamed the livers of these mice, whereas CSE, TA, RG, and AO administration alleviated these inflammatory response (Fig. 2j, k). IL-10 levels in liver homogenates of HFD-fed mice increased significantly after CSE, RG, and AO administration ($P < 0.05$, $P < 0.01$, $P < 0.001$) (Fig. 2l), suggesting CSE, RG, and AO promoted anti-inflammatory factor release, thereby reducing inflammatory responses. These data indicated that CS extracts and its bioactive compounds directly reduced liver transaminases and inflammation in NAFLD mice, and no significant difference was observed between CSE and TA.

*Cassiae Semen regulates gut microbiota imbalance in NAFLD mice.* Previous studies have shown that HFD caused gut microbiota dysbiosis in animals[3], with dysbacteriosis a common phenomenon in NAFLD pathogenesis. The V3-V4 region of 16 S rRNA were sequenced to analyze microorganisms in fresh feces, and to determine the effects of CS extracts and its components on gut microbiota composition in HFD-fed mice.

Alpha diversity analyses indicated that sequencing depth covered most of the diversity (Supplementary Fig. 1a–e). These results revealed that the richness and diversity of gut microbiota in HFD group mice were significantly reduced when compared with the control group ($P < 0.05$), but these levels recovered after CS extracts or compounds administration. OTU classification and taxonomic status identification results showed that OTUs in HFD group mice decreased when compared with the control group, and increased after the extracts and compounds were administered, suggesting that gut microbiota diversity increased (Supplementary Fig. 1f). Besides, taxonomic composition analyses showed that at the phylum level, the total relative abundance of Firmicutes and Bacteroidetes in the HFD group also decreased, and Proteobacteria increased ($P < 0.001$). However, after two extracts and two compounds were administered, the relative abundance in HFD-fed mice was reversed ($P < 0.05$, $P < 0.01$, $P < 0.001$) (Supplementary Fig. 1g). Thus, CS extracts and its components regulated gut microbiota disorder in NAFLD mice, and no significant difference was observed between CSE and TA.

*Cassiae Semen improves intestinal mucosal barrier injury in NAFLD mice.* Previous studies reported that a HFD increased intestinal permeability, and decreased the gene expression of tight

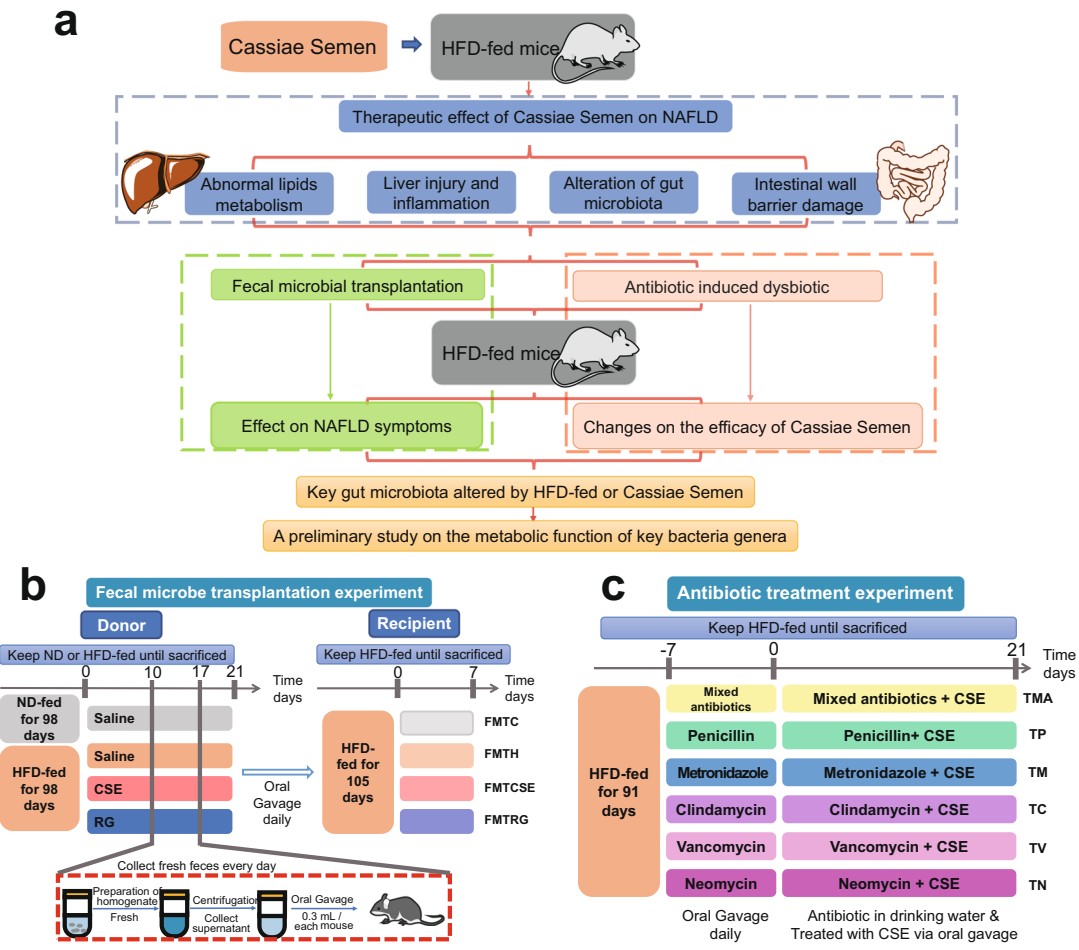

**Fig. 1 Research concepts and experimental design.** Three components studies were designed to verify our hypothesis (**a**). Experimental design of fecal microbe transplantation (FMT) (**b**) and antibiotic-induced dysbiosis studies (**c**). ND normal diet, CSE Cassiae Semen extract, RG Rubrofusarin-6-β-gentiobioside. FMTC, FMTH, FMTCSE, and FMTRG were the treatment groups receiving fecal microbial transplantation of mice in groups of control, HFD, CSE, and RG, respectively. TMA, TP, TM, TC, TV and TN denoted the groups treated with antibiotics mixture antibiotics, penicillin, metronidazole, clindamycin, vancomycin, and neomycin, respectively.

junction proteins, leading to metabolic endotoxemia [increased lipopolysaccharide (LPS) levels][19]. Increased serum LPS levels suggested intestinal barriers were damaged. After CSE, TA, RG, and AO were administered to HFD-fed mice, serum LPS levels in each group decreased ($P < 0.01$, $P < 0.001$) (Fig. 2m), suggesting metabolic endotoxemia was relieved. Intestinal mucosa-tight junction proteins, such as zonula occludens (ZO-1) and occludin play crucial roles in maintaining intestinal epithelial barriers. The results showed that CSE, TA, RG, and AO administration increased the relative expression of occludin and ZO-1 ($P < 0.01$, $P < 0.001$), indicating intestinal barrier integrity of HFD-fed mice was initially damaged, but was ameliorated after treatment (Fig. 2n, o). These data indicated that CSE, TA, RG, and AO improved the mechanical barrier of the intestinal mucosa, improved barrier integrity of the intestinal wall, and reduced metabolic endotoxemia.

**The hepatoprotective effects of Cassiae Semen are transferred by fecal microbiota transplantation.** To verify our hypothesis that the hepatoprotective effects of CS were related to the gut microbiota, an FMT strategy was designed. The fecal microorganisms of the control, HFD, CSE, and RG groups were transplanted to HFD-fed mice, and groups were named FMTC, FMTH, FMTCSE, FMTRG, respectively. Compared with the HFD group, TC, FFA, and LDL-C in each FMT group were

decreased in varying degrees (Fig. 3a, c, d). The liver transaminases (ALT, and AST), and the inflammatory factors (TNF, and IL-6) of the mice in each FMT group were found to be decreased compared to those of the mice in the HFD group (Fig. 3f–i), and the fecal microorganisms from the control group had a greater contribution in the increase of anti-inflammatory factors (IL-10) (Fig. 3j). Moreover, serum LPS levels were decreased in the FMTC, FMTCSE, and FMTRG groups (Fig. 3k), and the relative expression of occludin and ZO-1 mRNA was upregulated in the FMT groups, compared with HFD-fed mice (Fig. 3l, m). The gut microbiota diversity was increased after FMT, and the change in the microbisl abundance caused by HFD was basically recovered (Supplementary Fig. 2a, b). As shown in Supplementary Fig. 2c, compared with other FMT groups, the microbiota in the mice that received FMT from HFD-fed mice showed a slightly different trend. For example, the abundance of *Cupiravidus*, *Bilophila* and *Roseburia* increased only in the FMTH group. The abundance of *Dehalobacterium*, *Odoribater*, *Oscillospira*, and *Roseburia* increased in both the FMTC and FMTCSE groups, while there was no significant change the abundance of these genus in the FMTH group.

In general, the effects of CSE and RG on ameliorating lipid accumulation, liver injury, inflammation, and intestinal barrier damage in NAFLD mice were transferred via FMT, which contributed to our hypothesis that the hepatoprotective effect of

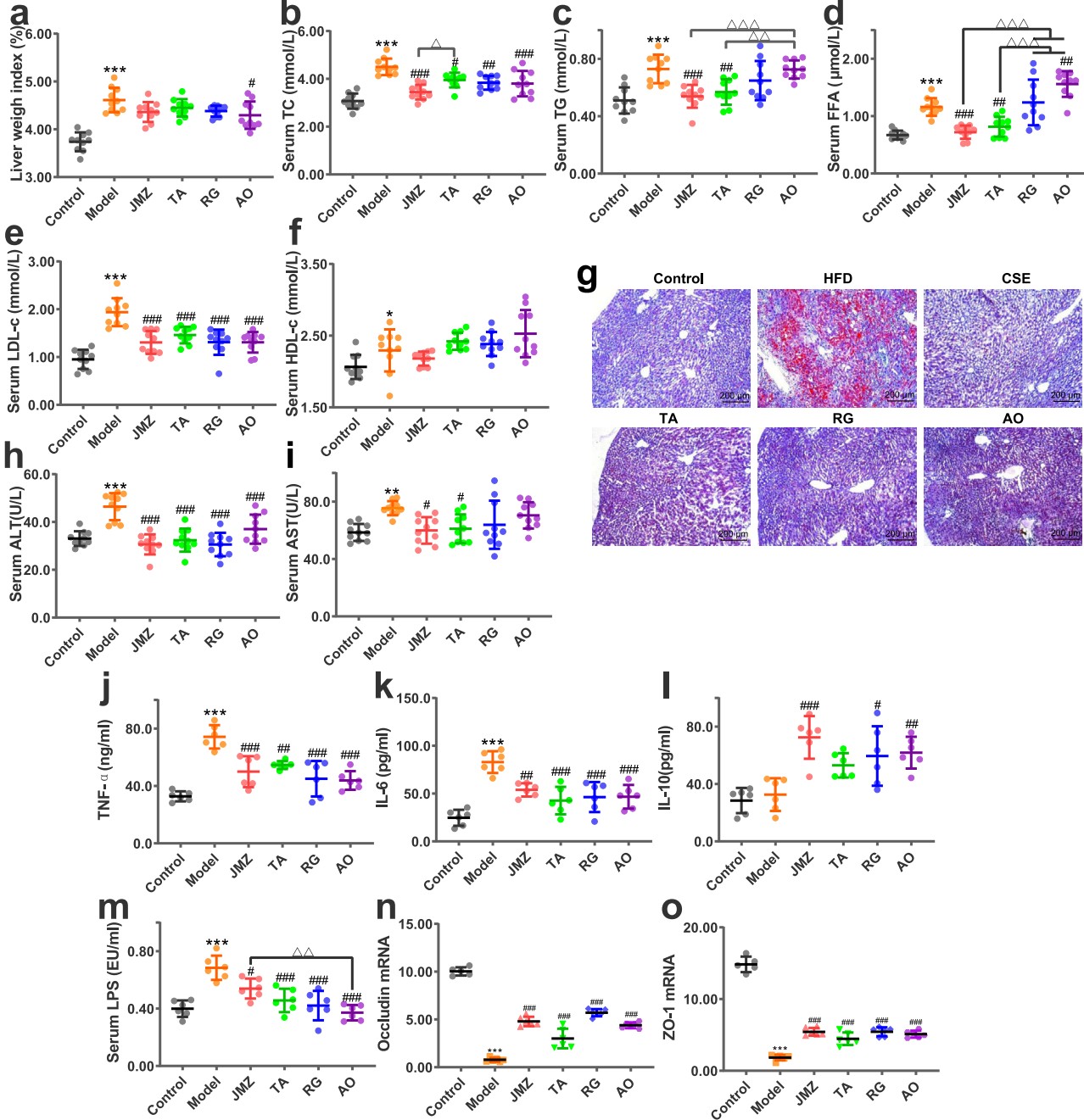

**Fig. 2 CSE (10 g/kg), total aglycone (TA) (10 g/kg), RG (20 mg/kg) and aurantio-obtusin (AO) (20 mg/kg) ameliorated liver weight indices and fat accumulation, liver injury and inflammation, intestinal mucosal barrier injury and altered microbiota diversity in HFD-fed mice.** Data for liver weight indices (**a**), serum total cholesterol (TC) (**b**), triglycerides (TG) (**c**), free fatty acids (FFA) (**d**), low-density lipoprotein cholesterol (LDL-c) (**e**), and high-density lipoprotein cholesterol (HDL-c) (**f**), $n = 10$ mice/group. Liver lipid accumulation. Liver tissue was stained with oil red O and observed under light microscopy (100×) (**g**). Alanine aminotransferase (ALT) (**h**) and aspartate aminotransferase (AST) (**i**) levels reflect the effects of HFD and administration on liver injury in mice, $n = 10$ mice/group. Tumor necrosis factor-alpha (TNF-α) (**j**), interleukin (IL)-6 (**k**), and IL-10 (**l**) levels reflect liver inflammation, $n = 6$ mice/group. Serum lipopolysaccharide (LPS) levels (**m**), occludin and zonula occludens (ZO) −1 mRNA relative expression (**n**, **o**), $n = 6$ mice/group. Data are presented as the mean ± standard deviation (SD). Statistical analyses were performed using Tukey's multiple comparison test. $*P < 0.05$, $**P < 0.01$, $***P < 0.001$ vs. control; $^{\#}P < 0.05$, $^{\#\#}P < 0.01$, $^{\#\#\#}P < 0.001$ vs. HFD; $^{\triangle}P < 0.05$, $^{\triangle\triangle}P < 0.01$, $^{\triangle\triangle\triangle}P < 0.001$.

CS has a strong correlation with gut microbiota. Fecal microorganisms in the control group also exerted the same effects on improving NAFLD. These observations suggested that normal gut microbiota structures of the control group, and the normalized gut microbiota structures of the CSE and RG groups somewhat alleviated abnormal lipid metabolism in NAFLD mice, and reduced liver damage and inflammation, thereby improving

intestinal barrier integrity. While FMTH group showed slightly similar results as the other three FMT groups. It was speculated that the reason was that, although the abundance of bacteria in the fecal microbial composition of the HFD group mice is low, but they still exist. These low beneficial bacterial levels may act as "primers" and exist in HFD-fed mice. Such beneficial bacteria generate synergistic effects and affect NAFLD processes via

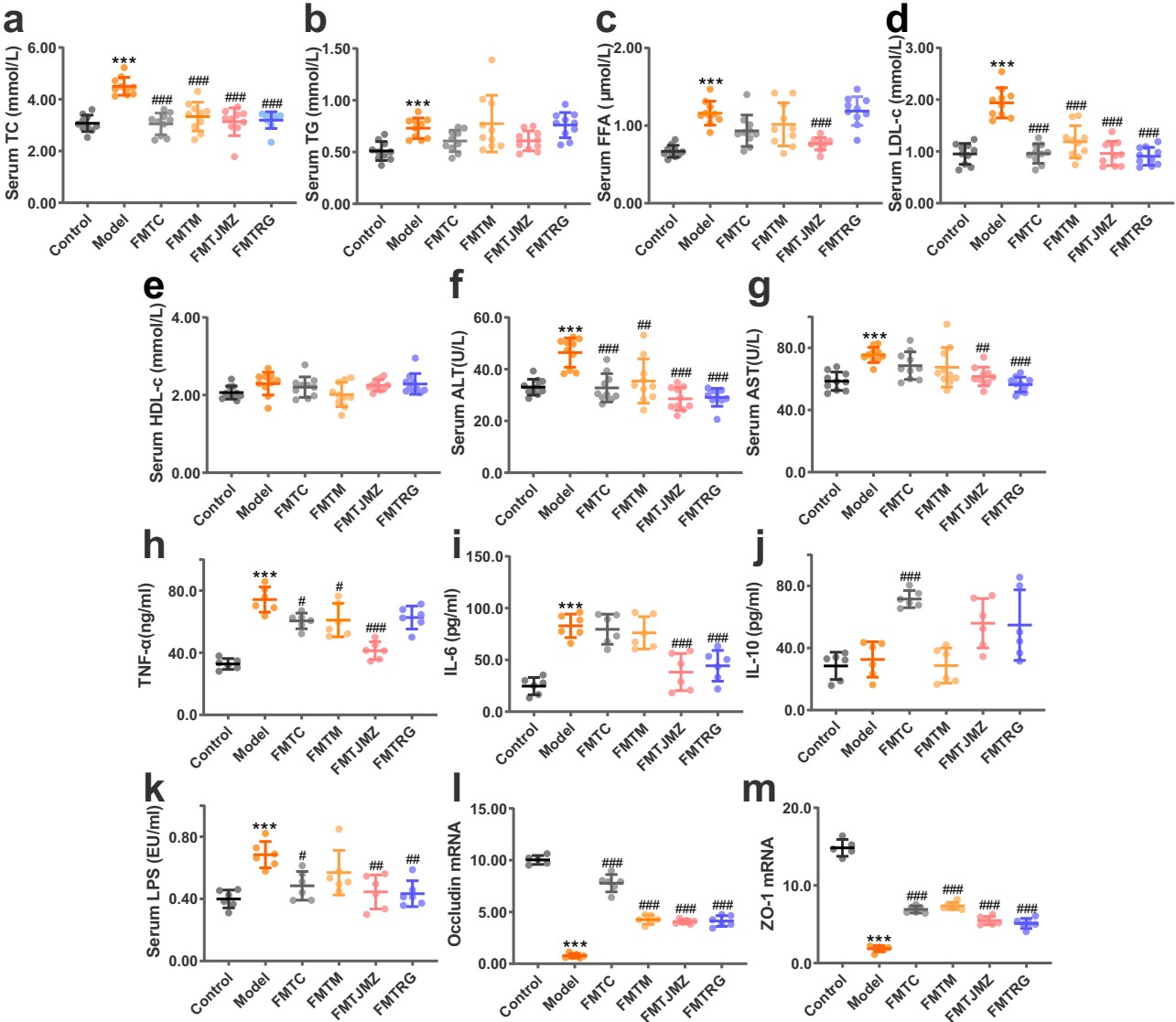

**Fig. 3 CS effects on fat accumulation, liver injury and inflammation, and intestinal mucosal barrier injury are transferred via FMT.** Serum TC (**a**), TG (**b**), FFA (**c**), LDL-c (**d**), HDL-c (**e**), Serum ALT (**f**), and serum AST (**g**), $n = 10$ mice/group. TNF-α (**h**), IL-6 (**i**), IL-10 (**j**), and serum LPS levels (**k**), relative expression of occludin (**l**), ZO-1 (**m**) mRNA levels, $n = 6$ mice/group. Data are presented as the mean ± SD. Statistical analyses were performed using Tukey's multiple comparison test. *$P < 0.05$, **$P < 0.01$, ***$P < 0.001$ vs. control; #$P < 0.05$, ##$P < 0.01$, ###$P < 0.001$ vs. HFD.

complex mechanisms. Our data suggested the colonization of gut microbiota in mice after FMT was highly complex, thus further research should be conduted to prove our hypothesis.

**Cassiae Semen effects are inhibited or eliminated by antibiotic-induced gut disorder.** It is accepted that antibiotics inhibit gut microbiota or induce dysbiosis. When the gut microbiota in HFD-fed mice was disturbed by antibiotics, the CS efficacy in NAFLD mice was weakened or eliminated, indicating that the complete structure of the microbiota is necessary for the full benefits of CS. To observe this, an antibiotic-induced gut disorder study was designed, and different types of antibiotics with different spectra were tested. HFD-fed mice were treated with penicillin, metronidazole, clindamycin, vancomycin, neomycin, or mixed antibiotics before CSE administration. At the time of CSE administration, HFD-fed mice were given free access to drinking water containing antibiotics to maintain the gut microbiota disorder. The labels, TMA, TP, TM, TC, TV, and TN were used to denote the groups treated with mixed antibiotics,

penicillin, metronidazole, clindamycin, vancomycin, and neomycin, respectively.

16 S rRNA sequences of gut microbiota from HFD-fed mice were investigated before and after antibiotic gavage treatment to observe the effects of oral antibiotics on gut microbiota. Compared with conditions before antibiotic treatment [the group without antibiotic treatment (NA)], gut microbiota diversity in HFD-fed mice was significantly reduced by antibiotic treatments, and the relative abundance of other bacteria was significantly decreased ($P < 0.05$) (Supplementary Fig. 3a, b, Fig. 4a). In general, HFD-fed mice had different degrees of dysbiosis after the oral administration of different antibiotics.

After antibiotics induced dysbiosis, CSE was administered to each mouse in the antibiotic treatment groups. Serum TC, and FFA levels were significantly higher than the CSE group after all these six antibiotic treatments ($P < 0.05$, $P < 0.01$, $P < 0.001$), and TC levels in the TMA group was even higher than the HFD group ($P < 0.001$) (Fig. 4b, d). TG in TM and TN groups, LDL-c in TMA and TV groups were increased than CSE group as well (Fig. 4c, e). Moreover, serum ALT and AST levels in the six antibiotic

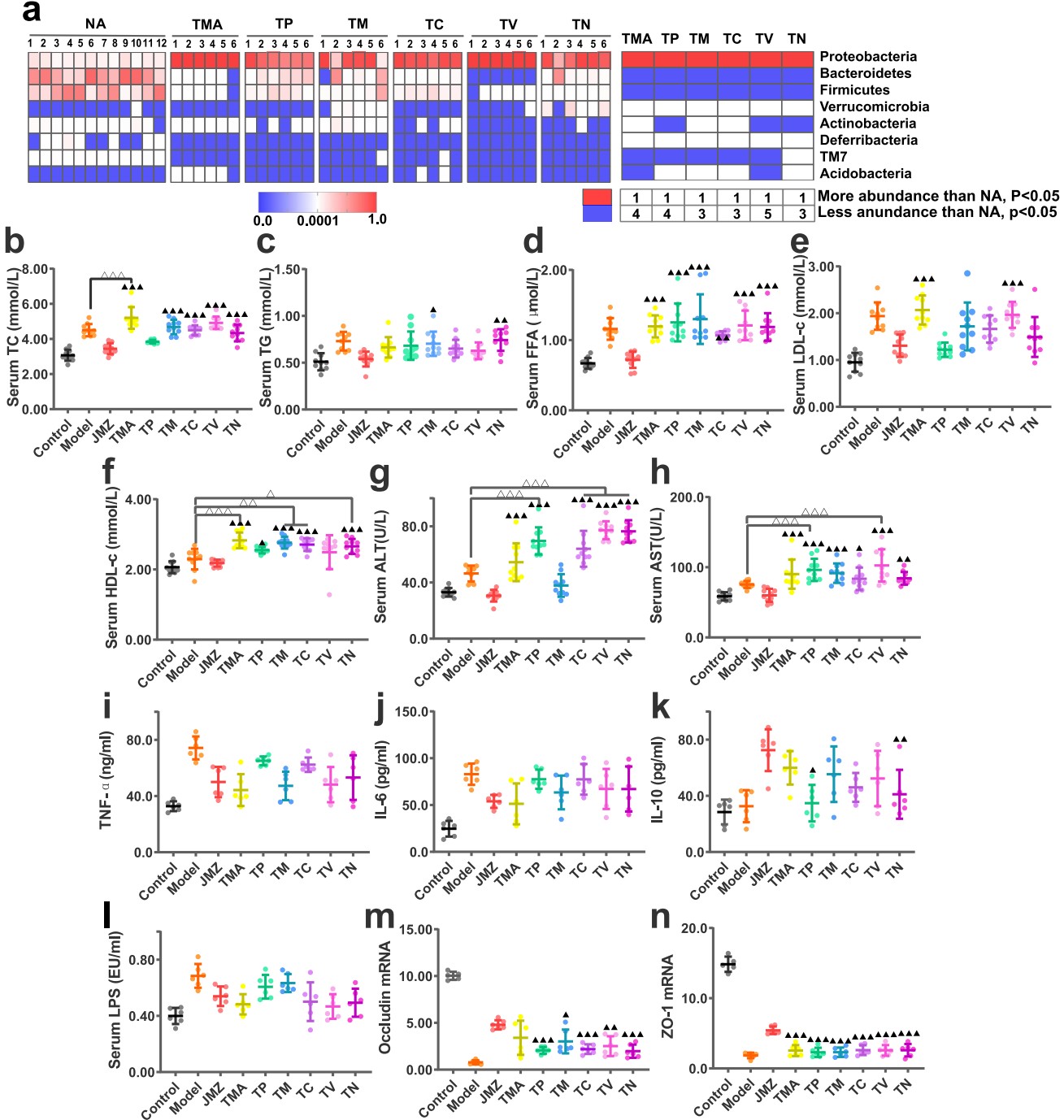

**Fig. 4 Extracts or compounds administration ameliorate antibiotic-induced dysbiosis in HFD-fed mice.** Phylum changes before and after antibiotic administration which before CSE administration (*n* = 6 each/group) (**a**). CS efficacy toward NAFLD was weakened or eliminated after HFD-fed mice gut microbiota was disturbed by antibiotics. Serum TC (**b**), TG (**c**), FFA (**d**), LDL-c (**e**), and HDL-c (**f**) levels. Serum ALT (**g**), serum AST (**h**) levels. Data are presented as the mean ± SD (*n* = 10 mice/group). TNF-α (**i**), IL-6 (**j**), IL-10 (**k**), and serum LPS (**l**) levels. The relative expression of occludin (**m**), ZO-1 (**n**) mRNA, *n* = 6 mice/group. Data are presented as the mean ± SD. Statistical analyses were performed using Tukey's multiple comparison test. ▲*P* < 0.05, ▲▲*P* < 0.01, ▲▲▲*P* < 0.001 vs. CSE group. ***P* < 0.001.

treatment groups were significantly higher than the CSE group, except for ALT in TM group (*P* < 0.05, *P* < 0.01, *P* < 0.001) (Fig. 4g, h). Of these, ALT levels in TP, TC, TV, TN groups, and AST levels in TP, TV groups were significantly higher than the HFD group (*P* < 0.001). Moreover, IL-10 levels in TP and TN groups were significantly lower than the CSE group (*P* < 0.05, *P* < 0.01) (Fig. 4k). When compared with the CSE group, no significant changes were observed in serum LPS levels in

antibiotic groups (*P* > 0.05) (Fig. 4l). This might be due to inhibited LPS production or decreased abundance of LPS producing bacteria after antibiotic treatment, meaning serum LPS levels were not significantly increased. The relative expression of occludin and ZO-1 mRNA in the ileum of the antibiotic groups was significantly lower than the CSE group (*P* < 0.05, *P* < 0.01, *P* < 0.001), except for occludin mRNA in the TMA group (*P* > 0.05) (Fig. 4m, n).

In general, gut microbiota disturbances induced by different antibiotics primarily inhibited the therapeutic effects of CSE toward lipid accumulation, liver injury and inflammation, gut microbiota imbalance, and intestinal barrier injury in NAFLD mice. Similarly, some antibiotics eliminated the effects of CSE. Thus, the therapeutic role of CSE was limited after gut microbiota structures were initially compromised by HFD, and then seriously damaged by antibiotics. Therefore, the therapeutic effects of CSE toward NAFLD depended on the complete structure of gut microbiota. These observations supported our hypothesis that the hepatoprotective effects of CSE were closely related to the gut microbiota.

**Key gut microbiota altered by Cassiae Semen during NAFLD treatment.** So far, our data indicated that CS alleviated liver fatty degeneration, abnormal lipid metabolism, liver injury, liver inflammation, dysbacteriosis, intestinal barrier damage, and enterogenous endotoxemia in NAFLD. These effects were presumably exerted by gut microbiota, and we try to identify these key bacteria through metastats analysis.

Metastats analysis was used to determine the bacterial structures significantly altered after HFD-fed and CS administration. Overall, HFD-fed and extract/compound administration altered a total of 39 genera when compared with the control group (Fig. 5a). The HFD group were enriched for 10 genera from Proteobacteria: *Sutterella* from Alcaligenaceae, *Janthinobacterium* from Oxalobacteraceae, and seven genera from Enterobacteria: *Enterobacter*, *Erwinia*, *Escherichia*, *Klebsiella*, *Morganella*, *Salmonella*, *Serratia*, and *Trabulsiella* ($P < 0.05$), and reversed after extract/compound administration, with CSE significantly reducing the abundance of *Erwinia*, *Klebsiella*, *Morganella*, and *Trabulsiella* ($P < 0.05$ and $Q < 0.05$) (Fig. 5a). In previous studies, these bacteria were associated with liver cirrhosis[20] and liver cancer[21]. The genera with reduced abundance in the HFD group (*Dehalobacterium* from Dehalobacteriaceae, *Adlercreutzia* from Coriobacteriaceae, *Odoribacter* from Odoribacteraceae, *Rikenella* from Rikenellaceae, *Mucispirillum* from ferribacteraceae, *Clostridium* from Clostridiaceae, *Dorea* and *Lachnobacterium* in Lachnospiraceae, and *Oscillospira* and *Ruminococcus* in Ruminococcaceae) increased after extract/compound administration ($P < 0.05$), especially *Dehalobacterium*, *Oscillospira*, and *Ruminococcus* after CSE treatment ($P < 0.05$ and $Q < 0.05$) (Fig. 5a).

In addition, *Coprococcus* and *Ruminococcus* genera from Lachnospiraceae were significantly increased after CSE treatment ($P < 0.05$ and $Q < 0.05$) (Fig. 5a). It was reported that bacteria from Ruminocaccae and Lachonospiraceae (*Blautia*, *Roseburia*) produced short chain fatty acids, which were reported to improve insulin resistance in NAFLD patients, prevent metabolic endotoxemia by strengthening gut barriers, and improve the chronic low-grade inflammatory state in NAFLD patients by promoting anti-inflammatory production, and regulating immune cell differentiation[22–24].

*Akkermansia* genera abundance was also increased after RG treatment ($P < 0.05$) (Fig. 5a). Studies have shown that *A. muciniphila* reverses a metabolic disorder induced by HFD, including increased fat mass, metabolic endotoxemia, inflammation of adipose tissue, insulin resistance, and promote the secretion of endogenous cannabinoids from the intestinal tract to control inflammation, intestinal barrier, and intestinal peptide secretion[25]. Thus, after our treatment regimen, those bacteria harmful to NAFLD processes were reduced and beneficial bacteria were enriched.

Our FMT data showed that fecal transplantation from the CSE group to HFD mice reduced the gut microbiota with increased abundance in HFD-fed mice, as well as increase the gut microbiota with decreased abundance in HFD-fed mice (Supplementary Fig. 2c). In general, FMT appeared to replicate the composition of the bacteria after CS extracts or compounds administration, and its therapeutic effects on NAFLD was reflected by changes in NAFLD indicators. These results further demonstrated that gut microbiota functions cannot be ignored during CSE treatment for NAFLD.

On the other hand, after the antibiotic treatment, the bacterial genera changed by the administration group was basically reversed (Supplementary Fig. 3c). After mixed antibiotics or vancomycin treatment, the abundance of bacteria decreased in the CSE group, increased ($P < 0.05$). From these trends, antibiotic treatments reduced the abundance of beneficial bacteria, and increased harmful bacteria for NAFLD. From the perspective of gut microbiota alterations, antibiotic treatments weakened or eliminated the therapeutic effect of CSE toward NAFLD. Moreover, CSE-mediated therapeutic effect were closely related with the gut microbiota.

To explore the putative role of these 39 key genera, Spearman correlation analysis was used to analyze correlations between these genera and NAFLD-related indicators. Our data indicated that genera enriched in the administration group were negatively correlated with blood lipid indices (TC, TG, FFA, and LDL-c), liver transaminases (ALT, and AST), inflammatory factor indices (TNF-α, and IL-6), while they were positively correlated with intestinal mucosa protein expression (Fig. 5b). Of these genera, *Odoribaracter* was negatively correlated with TC, HDL-c, and LDL-c levels, and positively correlated with occludin and ZO-1 expression; *Rikenella* was negatively correlated with TC and HDL-c levels, and positively correlated with ZO-1 expression. *Desulfovibrio* was positively correlated with occludin and ZO-1 expression. In contrast, bacteria enriched in the HFD group, *Enterobacter*, *Erwinia*, *Klebsiella*, *Salmonella*, *Serratia*, and *Trabulsiella* were positively correlated with TC levels; *Janthinobacterium*, *Enterobacter*, *Erwinia*, *Escherichia*, *Klebsiella*, *Morganella*, *Salmonella*, and *Trabulsiella* were positively correlated with LDL-c levels. Also, *Enterobacter*, *Erwinia*, *Klebsiella*, *Morganella*, *Serratia*, and *Trabulsiella* were negatively correlated with ZO-1 expression. *Klebsiella* and *Serratia* were negatively correlated with occludin expression. Thus, we hypothesized that enriched bacteria in the HFD group promoted NAFLD pathogenesis, especially *Janthinobacterium*, *Enterobacter*, *Erwinia*, *Escherichia*, *Klebsiella*, *Morganella*, *Salmonella*, *Serratia*, and *Trabulsiella*; and the enriched bacteria after CS extracts or compounds administration, alleviated fatty accumulation, liver injury, liver inflammation, and intestinal wall barrier disorders, predominantly included *Odoribacter*, *Rikenella*, and *Desulfovibrio*.

**A preliminary study on the metabolic function of key bacteria genera.** To further explore the role of these genera in NAFLD pathogenesis, microbial community functions were investigated using the Phylogenetic Investigation of Communities by Reconstruction of Unobserved States (PICRUST) algorithm. By comparing our 16 S rRNA gene sequencing data with a microbial function genome database (i.e., the KEGG database; http://www.genome.jp/kegg/pathway.html), we could predict bacterial metabolic functions[26]. During the prediction process, the copy number of 16 S rRNA genes from different species was considered, and species abundance data in the original data were corrected to ensure prediction results were accurate and reliable.

KOs is the gene identifier in Kyoto Encyclopedia of Gene and Genomes (KEGG) database, which is used to unify the same gene among different species. The unified gene KOs number of different species is the same. In the HFD group, a large number of KOs were significantly changed in the metabolism network

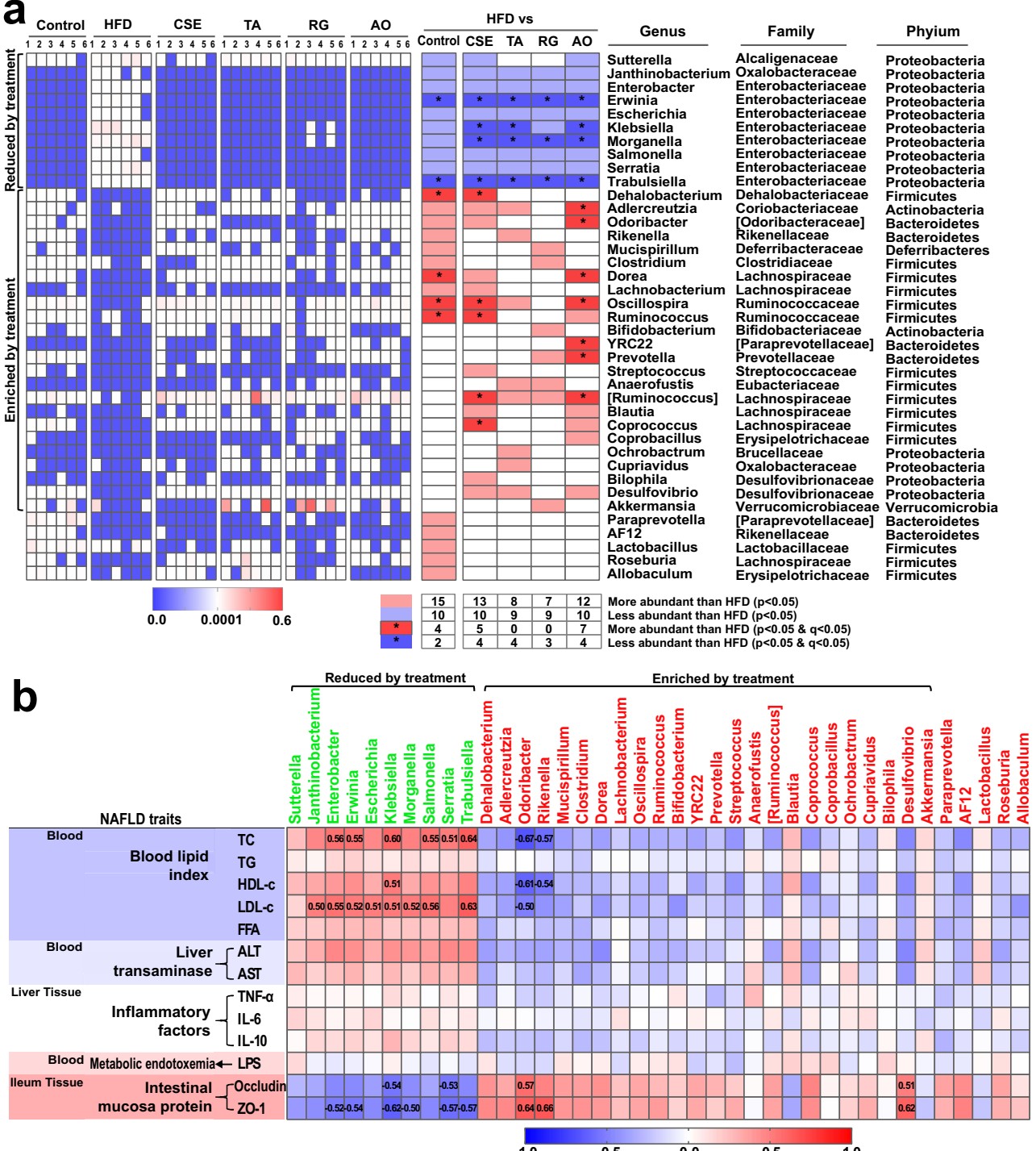

**Fig. 5 Genera changes by HFD and extracts or compounds administration.** Genera changes by HFD and extracts or compounds administration (**a**). Heatmaps on the left show that relative abundance of bacterial genera was altered. Statistical analysis of differences between groups was performed using Metastats; in the image, red represents increased relative abundance, blue represents decreased, and *represents a significant change in genus (both $P < 0.05$ and $Q < 0.05$). The corresponding genus information is on the right. Spearman correlations among 39 key genera and NAFLD-related indicators (**b**). Significant correlation coefficients between genera are indicated by the numbers in the boxes.

compared to the control, and we mainly analyzed those with a total relative abundance of more than 1‰, of which 884 were decreased and 1520 were increased (see online Supplementary Data 1). Among these, 2363 KOs were recovered after CSE treatment (875 enriched and 1488 reduced), 1965 KOs were recovered after TA treatment (727 enriched and 1238 reduced),

1758 were recovered after RG treatment (505 enriched and 1253 reduced), 2284 were recovered after AO treatment (818 enriched and 1466 reduced) (see online Supplementary Data 1). The bacterial metabolic pathways from the KEGG analysis were also screened to investigate which pathways were significantly changed by HFD or extracts/compounds administration.

Compared with the control group, 81 pathways were reduced in the HFD group, 80 pathways were recovered after CSE treatment, 76 pathways were enriched in the HFD group, and 75 pathways were decreased after CSE treatment; as well as 74, 67, and 80 increased and 57, 45, and 70 decreased in TA, RG, and AO, respectively (see online Supplementary Data 2). Previous studies have shown that HFD causes abnormal metabolism in fatty acids, bile acids and amino acids in animals[27]. From the PICRUST algorithm, which predicted the microflora metabolic functions, these abnormal metabolic pathways were investigated. When compared with the HFD group, 86 third-level functional pathways related to metabolism were observed have changed after CSE treatment, with 47 increased and 39 decreased (see online Supplementary Data 2). As shown in Fig. 6a, CSE mainly enhanced the lipid, carbohydrate, amino acid, and energy metabolism pathways, as well as cell motility in cellular processes, sporulation in cellular processes and signaling. After CSE treatment, the three-level metabolic function pathways related to primary and secondary bile acid biosynthesis and glycerolipid metabolism increased ($P < 0.0001$), while unsaturated fatty acid biosynthesis were decreased ($P < 0.0001$), which potentially explains the hypolipidemic effects of CSE (Fig. 6b). In addition, for the metabolic functional pathways related to polysaccharide biosynthesis and metabolism, CSE reduced the metabolic pathways with increased abundance in the HFD group, such as LPS biosynthesis, and LPS biosynthetic proteins ($P < 0.001$) (Fig. 6a). Meanwhile, CSE enriched functional pathways involved in cell motility and sporulation (many Firmicutes are motile and form endospores) (Fig. 6a).

KEGG differential genes ($Q < 0.05$) were screened and marked in the primary bile acid biosynthesis, glycerolipid metabolism, and lipopolysaccharide biosynthesis pathways (Fig. 6b and online Supplementary Figs. 4 and 5). The gene expression of choloylglycine hydrolase [EC:3.5.1.24] (K01442), was predicted to be upregulated in the biosynthesis of primary bile acid pathway after CSE treatment (Fig. 6b). Therefore, CSE may have the potential to promote primary bile acid biosynthesis, and the main metabolites might be taurine and/or chenodeoxycholate. Further experiments are needed to verify this. CSE treatment increased several genes in the glycerolipid metabolism pathway, including phosphate acyltransferase [EC:2.3.1.274] (K03621), and 1,3-propanediol dehydrogenase [EC:1.1.1.202] (K00086), indicating that CSE has the potential to promote glyceride metabolism (online Supplementary Fig. 4). Besides, HFD might upregulate the expression of a variety of genes in the LPS synthesis pathway, including *WaaL* (K02847), *WaaQ* (K02849), and *WaaA* (K02527), which participate in the synthesis of LPS from various *Escherichia coli* strains, while CSE treatment has the opposite effect on these enzymes (see online Supplementary Fig. 5), which consistent with the result that CSE could reduce the serum LPS of HFD-fed mice.

## Discussion

In Asia, CS is a commonly used Chinese herbal medicine. Its ability to reduce fevers, improve vision and loosen bowels to relieve constipation has been confirmed in research studies[6–9]. However, the relationship between CS effects on the liver and its effect on the gut microbiota has not been comprehensively investigated. Therefore, the HFD-fed mice were used to study the protective effect of CS toward liver, and assess correlations with gut microbiota. Our data indicated that CS prevented NAFLD by reducing fat accumulation, improving liver injury, reducing liver inflammation, regulating gut microbiota disorder, improving intestinal barrier injury, and metabolic endotoxemia. Of these indices, the effect of CSE on lipid accumulation was slightly

improved when compared with the other groups (TA, RG, and AO), which we hypothesized, reflected the integral therapeutic effect of TCMs. There was no significant difference between CSE and TA, which suggested that enzymatic hydrolysis of glycosides into aglycones in CS has similar effect with glycosides of CS on its hepatoprotective effect. Similarly, improvements in NAFLD symptoms via FMT of control groups transferred into HFD-fed mice were also demonstrated. Fecal microbiota transferred from mice treated with CSE into HFD-fed mice not only restored recipient gut microbiota composition to normal, but also reduced blood lipids, serum transaminases, liver inflammatory reactions, metabolic endotoxemia, increased intestinal mucosal protein expression, and improved intestinal barrier damage. Therefore, these FMTs relieved NAFLD symptoms, and CS's effect could be transferred to HFD-fed mice through FMT. In contrast, data from the antibiotic-induced gut microbiota disorder study indicated that the effects of CSE were weakened or eliminated by antibiotics. Antibiotic treatment partially eliminated the regulatory effects of CSE toward the gut microbiota. After antibiotic treatment, the abundance of gut microbiota in HFD-fed mice was reduced, and the gut microbiota structural disorder became more serious, suggested that the microbiota is necessary for the full benefits of CSE. These data indicated that the hepatoprotective effects of CS have a strong correlation with the complete structure of gut microbiota.

Several studies have reported that CSE have strong in vivo antibacterial effects, especially against common pathogens such as, Enterobacteriaceae and Enterococcus families from Proteobacteria, e.g., *Escherichia coli*, *Enterobacter aerogenes*, *Enterococcus faecalis*[28,29], *Staphylococcus aureus*, and *Clostridium perfringens*[30]. These in vitro data provided evidence that harmful Enterobacteriaceae were reduced in NAFLD mice after CS treatment. In addition, bacteria reductions during NAFLD were recovered; i.e., *Akkermansia*, *Allobaculum*, *Bifidobacterium*, *Blautia*, *Coprococcus*, *Coprobacillus*, *Desulfovibrio*, *Oscillospira*, *Prevotella*, and *Roseburia*[18,31–35] were all observed after CS treatment or FMT, suggesting positive roles in NAFLD relief. However, it was worth noting that for *Bilophila*, *Lactobacillus*, *Mucispirillum*, and *Paraprevotella*[18,31–33,36], their abundance increased during NAFLD, but decreased during our intervention. Correlations between key genera and NAFLD indices suggested that some genera enriched in HFDs were positively correlated with TC and LDL-c levels, but negatively correlated with occludin and ZO-1 expression. Thus, these genera may be involved in the exacerbations of NAFLD. In contrast, genera enriched by CS, especially *Odoribacter*, *Rikenella*, and *Desulfovibrio*, were beneficial for NAFLD.

Gut microbiota functional predictions also highlighted a regulatory bacterial biosynthetic potential for primary and secondary bile acid and metabolism of glycerolipid by CSE treatment, indicated that gut microbiota, modulated by CSE, may promote lipid digestion and absorption, reduced lipid accumulation in HFD-fed mice, and alleviated lipotoxicity. Moreover, increasing pathway in biosynthesis of LPS corresponded to decreased Proteobacteria abundance, as it is a major component of the outer membrane of Gram-negative bacteria, and LPS biosynthesis is significantly correlated with liver steatosis[37]. The enriched function pathways in cell motility and sporulation corresponded to the increasing Firmicutes abundance after CSE treatment. Thus, CS extracts and its components potentially regulated the abundance and composition of intestinal microorganisms with specific metabolic functions, thereby improving metabolic endotoxemia and lipid accumulation in NAFLD mice. These might be manifested as reductions in intestinal barrier damage, liver injury and inflammatory response caused by lipid toxicity, bacterial translocation, and other factors.

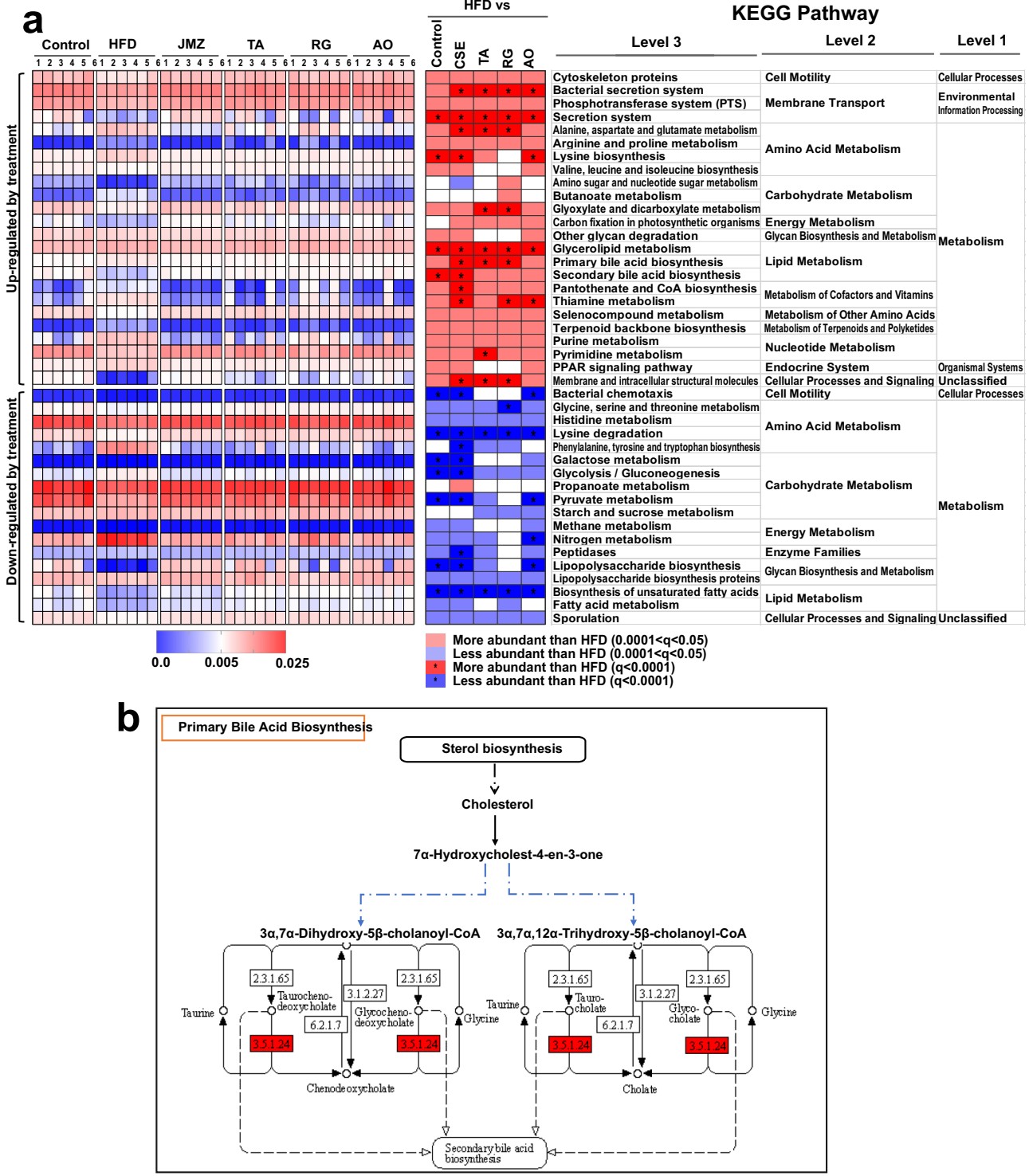

**Fig. 6 KEGG pathway analysis.** Metabolic pathway with significant enhancement by CSE treatment (**a**). The heatmaps on the left show that the relative abundance of several metabolic pathways was altered. Statistical analysis of the differences between the groups was performed using the original false discovery rate method of Benjamini and Hochberg. In the image, red represents increased relative abundance, blue represents decreased relative abundance, and * represents $Q < 0.0001$. The corresponding genus information is on the right. Effects of CSE on the primary bile acid biosynthesis pathway (**b**).

In conclusion, our study suggested that the extracts and compounds of CS reduced lipid accumulation, liver injury and inflammation, gut microbiota disorders, intestinal barrier injury, and metabolic endotoxemia in HFD induced liver injury. Our FMT and antibiotic studies supported the notion that CS effects were related to gut microbiota, i.e., reduced lipids and improved endogenous endotoxemia by potentially altering gut microbiota with specific metabolic functions so as to reduce intestinal barrier damage and liver inflammation, and alleviating NAFLD symptoms. Our comprehensive analyses have characterized some of the gut microbiota implicated in NAFLD, and importantly, as well as the hepatoprotective mechanisms of CS for NAFLD. Our

study also provided a feasible way to explore the mechanism of traditional Chinese medicine treatment of diseases.

## Methods

**Preparation of administrated drugs.** CS was purchased from Bozhou market (Anhui, China) and authenticated by Professor Zhuju Wang. To prepare CSE, 50 g CS dried powder was extracted under reflux in 500 ml of 75% ethanol for 2 h, and then filtered. The filtrate was evaporated until dry, and the dense extract was dissolved in water to generate a 50 ml extract (1 g ml$^{-1}$) for oral administration. To prepare TA, 50 g CS dried powder was added to 20 ml water to generate a paste. The solution was maintained at a constant temperature of 40 °C for 1 h, then ethanol was added to generate 75% ethanol in 500 ml. The solution was heated and refluxed for 2 h, then filtered. The filtrate was treated as before to generate TA extracts (1 g ml$^{-1}$). RG and AO are isolated and purified using modern chromatographic methods and repeated column chromatography, and determined by nuclear magnetic resonance[38,39]. Compound purity was determined at ≥98% by high performance liquid chromatography, using the normalization peak area method. RG and AO were separately added to sterile water to prepare a 2 mg ml$^{-1}$ solution for oral administration. The dosages of extracts and compounds were determined in pilot experiments.

**Animal experiments.** C57BL/6 male mice (18–22 g, six-weeks-old) were purchased from SPF Biotechnology Co., Ltd (Beijing, China), and maintained under specific-pathogen-free conditions. Animals were housed in a controlled environment (12 h/light–dark cycle at 20–22 °C and 45 ± 5% humidity), with free access to food and water. All animal studies were approved and performed in accordance with guidelines from the Institutional Animal Care and Use Committee of China Academy of Chinese Medical Sciences (Beijing, China). The project identification code was 20192005.

In total, 160 C57BL/6 mice were used, 60 for the pharmacological experiments, 40 for the FMT experiments, and 60 for the antibiotic-induced bacterial dysbiosis experiments. The control group (10 mice) was fed a normal diet (12.0% energy from fat, Beijing KeAo XieLi Company, Ltd., Beijing, China). In total, 150 mice were fed a HFD (15% lard, 2.5% cholesterol, 0.5% bile salts, 15% dextrin, and 67% basal feed, Xiaoshu Youtai (Beijing) Biotechnology Co., Ltd, Beijing, China) for 17 weeks to establish an NAFLD model. At the end of week 13, 50 HFD-fed mice were randomly divided into five groups (10 mice/group) for pharmacological studies. From week 14, control and HFD group mice were supplemented daily with 300 μL sterile saline. The administration groups were given: CSE (10 g kg$^{-1}$), TA (10 g kg$^{-1}$), RG (20 mg kg$^{-1}$), and AO (20 mg kg$^{-1}$) by intragastric gavage once a day. After three weeks (21 days) treatment, whole bloods were collected from the ophthalmic venous plexus, and serum was separated for serum marker quantification. Mice were sacrificed by cervical spondylolisthesis, with liver and ileum tissue extracted. The liver was weighed and the liver weight index calculated (liver weight index = liver weight/body weight × 100%). Part of the liver tissue was stored in 4% polyformaldehyde solution for pathological investigation, and the remaining liver and ileum samples were immersed in liquid nitrogen and stored at −80 °C for subsequent analysis. The FMT and antibiotic-induced dysbiosis studies are described later.

**Liver tissue histology.** Liver tissues were fixed in 4% paraformaldehyde, rapidly dehydrated and sliced into 6 μm thick slices using a frozen microtome (Thermo, USA). Frozen liver sections were stained with oil red for 25 min and then re-stained in hematoxylin for 5 min. When liver sections were examined under light microscopy (LEICA, USA), lipid droplets inside cells appeared red.

**Serum marker and endotoxin quantification.** Serum transaminases included: ALT and AST; Lipid indicators included: TG, TC, HDL-C, and LDL-C; FFA were detected using an automatic biochemical analyzer (BECKMAN COULTER AU480, US), according to the manufacturer's protocol. The serum endotoxin (LPS) was quantified using a tachypleus amebocyte lysate kit (Xiamen Bioendo Technology Co., Ltd, China) according to manufacturer's instructions.

**Measurement of hepatic inflammatory factors.** Liver tissues were weighed, to which 0.9% normal saline was added according to the weight proportion (g): volume (ml) = 1:9. A 10% tissue homogenate was generated using a LANYI-DLM Various Refrigeration Grinding Machine (10000 rpm, Shanghai Lanyi Industrial Co., Ltd., China). The supernatant was centrifuged for 10 min (2500 rpm), then assayed. Enzyme-linked immunosorbent assay kits (Beijing North Institute of Biotechnology Co., Ltd, China) were used to measure TNF-α, IL-6, and IL-10 serum levels, according to the manufacturer's instructions.

**Measurement of intestinal mucosa-tight junction proteins.** To investigate intestinal barrier integrity, the relative mRNA expression levels of two intestinal mucosa-tight junction proteins, occludin and ZO-1 from ileum tissue, were quantified using quantitative real-time polymerase chain reaction (qPCR). First, ileum tissues were ground with liquid nitrogen, then total RNA was extracted using Trizol Reagent (Invitrogen, USA). Total RNA (200 ng) from each sample were reverse transcribed to form the cDNA templates using a SuperScript III RT Kit

(ABI-Invitrogen, Invitrogen, USA). The qPCR primers were designed and synthesized (MDL biotech Co., Ltd, China) and the forward and reverse sequences are shown in Supplementary Table 1. The reaction was run in an ABI 7900Fast Real-Time qPCR System (Applied Biosystems, Thermo Fisher Scientific, USA). All the procedures were performed following the manufacturer's instructions. Relative mRNA expression was calculated using the $2^{-\Delta\Delta Ct}$ method.

**Fecal microbial transplantation.** Feces from the control, HFD, CSE, and RG groups were transplanted into HFD-fed mice to observe whether CSE and RG efficacy could be transferred via FMT. In FMT donors, the CSE group represented the extract administration group, and RG represented the compound administration group.

Our FMT approach was based on a previous study[40], but was modified and improved. The control, HFD, CSE, and RG groups served as FMT donors, and after 10 days of continuous administration, their feces were collected for FMT. Recipient mice (40 mice) had been fed a HFD for 15 weeks (105 days). They were randomly divided into four groups with 10 mice/group. Each group received FMTs from control, HFD, CSE, and RG group mice for seven days, respectively. Groups were renamed FMTC, FMTH, FMTCSE, and FMTRG groups.

To prepare fecal microbiota, fresh fecal pellets were collected daily into sterile tubes, and homogenized in sterile saline. Approximately 1 ml saline was used for every 100 mg feces. After centrifugation at 2000 rpm at 4 °C for 1 min, bacteria-enriched supernatants were collected and administered as soon as possible to relevant HFD-fed mice by gavage (0.1 ml per 10 g).

**Antibiotic-induced gut disorder.** At the beginning of week 13, 60 HFD-fed mice (10 mice/group) were orally administered 150 mg kg$^{-1}$ penicillin, 150 mg kg$^{-1}$ metronidazole, 150 mg kg$^{-1}$ clindamycin, 100 mg kg$^{-1}$ vancomycin, 200 mg kg$^{-1}$ neomycin or a mixture of the above (mixture antibiotics: 50 mg k$^{-1}$g penicillin + 60 mg kg$^{-1}$ metronidazole + 50 mg kg$^{-1}$ clindamycin + 25 mg kg$^{-1}$ vancomycin + 60 mg kg$^{-1}$ neomycin) once a day for 7 consecutive days. After the main antibiotic treatment (at week 14 of the HFD-fed), mice were treated with CSE by oral gavage for three weeks (21 days) (groups were renamed TP, TM, TC, TV, TN, and TMA), similar to the administration group, and antibiotics were maintained in drinking water to sustain gut microbiota dysbiosis. Drinking water contained 0.5 mg ml$^{-1}$ penicillin, 0.5 mg ml$^{-1}$ metronidazole, 0.5 mg ml$^{-1}$ clindamycin, 0.25 mg ml$^{-1}$ vancomycin, 0.5 mg ml$^{-1}$ neomycin, or mixture antibiotics (0.2 mg ml$^{-1}$ penicillin + 0.2 mg ml$^{-1}$ metronidazole + 0.2 mg ml$^{-1}$ clindamycin + 0.1 mg ml$^{-1}$ vancomycin + 0.2 mg ml$^{-1}$ neomycin).

**Gut microbiota analysis.** Fecal samples were collected at day one before blood collection, and snap-frozen in liquid nitrogen before storage at −80 °C. The 16 S rRNA V3-V4 hypervariable region from fecal samples was subjected to high-throughput sequencing by Shanghai Personal Biotechnology, Co., Ltd (Shanghai, China). Total bacterial genomic DNA was extracted using the Fast DNA SPIN extraction kit (MP Biomedicals, Santa Ana, CA, USA). DNA was quantified and visualized using a NanoDrop ND-1000 Spectrophotometer (Thermo Fisher Scientific, Waltham, MA, USA) and agarose gel electrophoresis, respectively. The V3–V4 region was PCR amplified using a forward (5′-ACTCCTACGGGAGG-CAGCA-3′) and reverse primer (5′-GGACTACHVGGGTWTCTAAT-3′). Sample-specific 7-base pair (bp) barcodes were incorporated into the primers for multiplex sequencing. PCR amplicons were purified using Agencourt AMPure beads (Beckman Coulter, Indianapolis, IN, USA), and quantified using the PicoGreen dsDNA assay kit (Invitrogen, Carlsbad, CA, USA). Amplicons were pooled in equal amounts after individual quantification steps, and pair-end 2 × 300 bp sequencing was performed on the Illlumina MiSeq platform, using a MiSeq Reagent Kit v3.

The Quantitative Insights Into Microbial Ecology (QIIME, v1.8.0) pipeline was used to process sequencing data. After disqualified and chimeric sequences were removed, the remaining high-quality sequences were clustered into operational taxonomic units (OTUs) at 97% sequence identity, using UCLUST (https://drive5.com/usearch/manual/uclust_algo.html). A representative sequence was selected from each OTU using default parameters. OTU taxonomic classification was conducted using BLAST (https://blast.ncbi.nlm.nih.gov/) searching the representative sequences set against the Greengenes Database (http://greengenes.secondgenome.com/)[41] using the best hit[42].

**Statistics and reproducibility.** Data from more than three experimental replicates were expressed as the mean ± SD. Tukey's multiple comparison test was used for multivariate comparison. Statistical analyses were performed using GraphPad Prism V.8.0 (GraphPad Software, USA), and differences were considered statistically significant at $P < 0.05$. *$P < 0.05$, **$P < 0.01$, ***$P < 0.001$ vs. control; #$P < 0.05$, ##$P < 0.01$, ###$P < 0.001$ vs. HFD. ▲$P < 0.05$, ▲▲$P < 0.01$, ▲▲▲$P < 0.001$ vs. CSE group.

Histograms and heat maps were drawn using GraphPad Prism V.8.0 (GraphPad Software, USA). 16 S rRNA sequence data analysis of gut microbiota and OTU analysis was performed using QIIME and R packages (v 3.2.0). Alpha diversity analyses, such as Rarefaction curves and Shannon index analyses were calculated using the OTU table in QIIME. Beta diversity analyses were performed to investigate structural variations in microbial communities across samples using

Unifrac distance metrics[43], visualized by principal coordinate analysis, and the unweighted pair-group method using arithmetic means hierarchical clustering[44]. Differences in Unifrac distances for pairwise comparisons among groups were determined using Student's $t$ test and the Monte Carlo permutation test with 1000 permutations, and visualized using box-and-whiskers plots. Taxa abundance at the phylum and genus levels were statistically compared between groups using Metastats[45]. Bacterial genera showing statistically significant differences by Metastats were also evaluated using the non-parametric Kruskal–Wallis test, with false discovery rate correction for multiple testing ($Q < 0.05$). Correlation coefficients between bacterial genera and NAFLD traits were determined using Spearman's correlation analysis. Bacterial metagenome content was predicted from 16 S rRNA gene-based microbial composition, and functional inferences were determined using the KEGG catalog, using the PICRUSt algorithm[26]. The original false discovery rate method of Benjamini and Hochberg was used to correct the multiple comparisons of KOs and metabolic pathways.

**Reporting summary**. Further information on research design is available in the Nature Research Reporting Summary linked to this article.

## Data availability

Raw 16 S rRNA gene sequence data for the feces microbiota were deposited in the NCBI Short Read Archive under BioProject Accession Number PRJNA715931. The source data for all figures are provided in Supplementary Data 3–7. The authors declare that all other data supporting the findings of this study are available within the article and its supplementary information files.

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

## Acknowledgements

This work was supported by the Scientific and technological innovation project of China Academy of Chinese Medical Sciences (CI2021A04202, CI2021A05208), Youth Fund Project of National Natural Science Foundation of China (No. 81803707), National Natural Science Foundation of China (No. 81673602), National Administration of Traditional Chinese Medicine of China (ZZ13-019).

## Author contributions

H.Y.L., H.W.W. and L.Y.T. contributed to the conception or design of the work; H.Y.L. searched the literature, collected the data, and drafted the manuscript; H.Y.L., S.M.X. and L.Y.T. contributed to the acquisition, analysis, and interpretation of data; Y.Q.L., C.L., L.X.W., X.K.Y. and X.Z. helped with conducting animal experiments and related tests; S.M.X., L.Y.T. and Z.J.W. contributed comments for version of the manuscript. All authors read and approved the final manuscript.

## Competing interests

The authors declare no competing interests.
