## [Transparent Peer Review File · Communications Biology]

Reviewers' Comments:

Reviewer #1:

Remarks to the Author:

The authors provide results of an interesting study wherein mice subjected to long-term high-fat diet (HFD) were used as a model of non-alcoholic fatty liver disease (NAFLD). Mice were allowed to develop NAFLD, and then administered treatments consisting of the traditional Chinese medicine (TCM) herb *Cassiae semen*, or specific extracts from the herb. Outcome measures included serum markers of dyslipidemia, serum ALT and AST as markers of hepatic injury, serum LPS and mRNA levels of two tight junction proteins as markers of gut permeability, a handful of cytokines, and 16S rRNA amplicon sequencing data to assess the fecal microbiota. Following description of the phenotype in HFD-fed mice with and without the herb compounds, the authors perform FMT and antibiotic exposure experiments to support a microbiota-dependent effect of *Cassiae semen*. While the data strongly support a GM-mediated effect, the exact nature of that effect is still purely correlative. In other words, there are no data presented showing a specific effect of one bacterial taxon or metabolite on NAFLD progression. Rather, the data show nicely that there are multiple features of NAFLD that are reversed (to varying degrees by the tested compounds), and present a wide array of potential changes in microbiota composition and function occurring simultaneously during HFD-induced NAFLD. As such, my biggest overall comment is for authors to review and revise the text critically along these lines, being careful to ascribe too much causality to specific taxa or predicted metabolites.

Major

1. Figure 1 is overly busy and includes information that should be reserved for Results. An example of excessive imagery includes the solid orange horizontal lines which really add nothing. Text is also excessive in spots, and the figure would be much improved if it contained only the specific experimental treatments and assays performed, without outcomes and descriptions of the treatments. Similarly, panels B and C are filled with abbreviations that have not been addressed in the manuscript yet. All of these should be spelled out in the figure legend.
2. Related to comment 1, all of these compounds need to be defined at their first usage in the Results (lines 57), and described for the reader. The results describe the effects of these abbreviated compound names, and it took me several minutes to deduce that JMZ was actually the plant (referred to as *Cassiae semen* in the title so this was unintuitive), as well as the identity of the other compounds being tested. The Introduction should clearly state how and why these specific compounds were selected for testing.
3. Figure panels need to be described in order in the text. Here, panel 2L (line 67) is followed by reference to panel 2P (line 81) and 2Q (line 84). Figure 2M is then presented in line 90.
4. The FMT data are somewhat problematic, and I'm not sure they convincingly show a specific GM-dependent effect. While the FMT recipient receiving material from control mice appear to be protected in terms of many outcomes, the fact that mice receiving material from 'model' mice are also protected is not adequately addressed. Figures 3N and O show that those mice (group FMTM) are protected from the HFD-induced dysbiosis, yet they should be receiving material similar to their existing HFD-induced microbiome, so why do these mice look like control mice? The authors focus on the beneficial effects of the compounds being tested and seem to ignore that transfer of the model (i.e., dysbiotic) microbiota was also protective.
5. The entire antibiotics portion of the Results is frustrating to read, as it's impossible to discern what the groups are without referring to the Methods. In fact, the Results don't even directly state that different types of antibiotics with different spectra were tested. In contrast, there is material in the Methods that is more appropriate in the Introduction or Results (e.g., lines 433-436). Methods should be limited to descriptions of what was done, while descriptions of the rationale and interpretation of tests and experiments belong in the Introduction or Results. The authors are advised to revise the text from the perspective of an average reader that often reads the abstract, and then the introduction and results for further details. Without reading the Methods, can a reader understand what is being done (and why) in each section of the Results?
6. The portion of the Results speculating on the causal relationships between specific bacteria and NAFLD should be removed or, at a minimum, moved to the Discussion. While the authors make a good

case for the microbiota being at least partially involved in the effects of Cassiae semen, the correlations shown in Figure 5B are just associations, in a largely dichotomous system (i.e., 'model' vs all other groups in Fig 5A).

7. Figure 6 is kind of a self-fulfilling prophecy in that you're showing the correlations between changes in composition and predicted pathways (made using those same changes in composition). Isn't it to be expected that these outcomes would agree very well? Also, the final statement of the Results is not specifically supported by any data shown here. No metabolites were actually measured and metabolic pathways are inferred using PICRUSt software which is not necessarily the most accurate tool. The same comments apply to the sentence on lines 327-8. These data do not make the predictions any more credible, and the portion of the Discussion describing these correlations over-states the implication of these findings.

8. Line 306 over-interprets the data, as the data do not definitively show that the hepatoprotective effects of Cassiae semen were due to "improving gut microbiota structures". While the antibiotic experiments suggest that the microbiota is necessary for the full benefits of Cassiae semen, it's no conclusive that the fully restored structure is required. The same can be said of the sentence at the start of line 325. It cannot be stated definitively that these genera exacerbated NAFLD, based on these data. Correlations do not necessitate causation.

Minor

1. On line 32, the Introduction states that Cassiae semen (CS) is used "to relieve constipation, hypolipidemia..." Should this be "hyperlipidemia", as CS reduces fat accumulation?

2. NAFLD should be defined at its first usage in the abstract and main text (line 34).

3. The word "flora" (line 42) is somewhat archaic. Consider changing "flora disorder" (same line, also on lines 140, 142, and in Figure 1) to "dysbiosis".

4. For the sake of clarity, I recommend changing "model" to "HFD" throughout (lines 77, 79, 82, etc.) and figures. The word "model" is too vague.

5. Recommend changing the word "executed" in Figure 1 (and anywhere else it appears) to "sacrificed" or "euthanized".

6. Suggest adding to line 56 so as to read, "...the effects of Cassiae semen and several of its bioactive compounds on hepatic..."

7. At first usage, please define host compounds being tested (line 58).

8. What is meant by 'monomers' (line 61)? Presumably this refers to RG and AO, but this should be defined for readers (unless I missed it).

9. Remove "And" as first word in sentences beginning on line 64 and line 145. In general, sentences should never begin with a conjunction (e.g., and, or, but).

10. Correct "occluding" (line 93).

11. Authors are advised to overstate the "successful establishment of our antibiotic-induced flora disorder animal model" (lines 140-141, 164). The fact that antibiotics alter the GM isn't necessarily model validation - what is being modeled? Changes in human GM exposed to same antibiotics?

12. Figure 4 is confusing in its use of NA, MA, P, M, etc. for group names in panels A to C, and then switches to different names in the remaining panels. Again, after searching through the Methods, I realized that these are the same groups, just at different time-points. Use a uniform set of group names and specify the differences in the figure legend regarding timing of sample collection.

13. Authors are advised to remove the word "pharmacodynamics" from line 161 (and elsewhere) as there are no pharmacodynamics data presented here.

14. Posing a question to the reader (line 176) is overly colloquial.

15. What is "metastats analysis" (line 177)?

16. Sentence beginning on line 251 belongs in Discussion. Same with sentences from lines 264 to 267, and ending on line 273.

17. The authors are advised to cite the primary literature showing these beneficial effects (line 294).

Reviewer #2:

Remarks to the Author:

General comments

This work mainly studied the effects of the traditional Chinese medicine "Jue Ming Zi"(Cassiae Semen, seeds of *Cassia obtusifolia* L. and *C. tora* L) on the hepatoprotective effects via modulation of gut microbiota. According to the authors, Cassiae Semen extract has positive influences on the gut-liver axis, which could produce beneficial metabolites and reduce harmful metabolites. The authors used 16s rDNA sequencing for microbiota compositional analysis and also fecal microbiota transplantation to check whether FMT could transfer the pharmacological effect of Cassiae Semen to HFD-fed mice. The authors made great efforts on the topic and many data were generated. However, the reviewers find that the whole story lacks of integration and the results part is not thoroughly discussed. In addition, the discussion part lacks of comprehensive analysis. As for the statistical methods used in the study, it is not sufficient. Language issue is also a major problem. Please consider using professional language editing service or ask native English speakers to improve the manuscript. In sum, the reviewer suggests major revision before the manuscript could be considered for publication on Communications Biology.

Minor issues

1. Line 6. No verb in the sentence of "Cassiae Semen, seeds of *Cassia obtusifolia* L. and *C. tora* L, so-called "Jue-Ming-Zi" in Chinese." Please revise the sentence into "Cassiae Semen, seeds of *Cassia obtusifolia* L. and *C. tora* L, is also known as "Jue-Ming-Zi" in Chinese."
2. Line 7. "Cassiae semen relieves constipation and exerts hepatoprotective effects" Please pay attention to third-person singular.
3. Line 10. "We examined the relationships... and mechanisms..."
4. Line 12 Our data... inflammation, hence having a good protective effect on"
5. When using the abbreviation NAFLD for the first time, please give the full name of the abbreviation. For example, "Non-alcoholic fatty liver disease (NAFLD)"
6. Line 21 "Consumption of ...with fat accumulating, increasing the risk of...". It is not correct to use "increased the risk"
7. Line 22-24: verb is missing in the sentence. Please check the manuscript and revise this type of grammar mistake. Please also consider using professional language editing service for the manuscript.
8. Line 30-31: please delete "in China" due to the repetition.
9. Line 44: The experimental setup is shown in Fig. 1. Please remove the brackets ().
10. There are many abbreviations used in the figure 1. Could the authors add the full term-abbreviation pair in the figure legend. For example, HFD:high fat diet, NAFLD: non-alcoholic fatty liver disease...
11. Line 56. "C57BL/6 mice were fed with HFD for..."
12. What is JMZ, TA, RG, and AO? When using the abbreviations for the first time, please give the full terms.
13. Line 78. After administration of "what"?
14. Line 124: please make sure that the tenses in the same sentence are the same. "Our data suggested the colonization of gut microbiota in mice after FMT was highly complex. Thus, more

research should be required to prove our hypothesis".

15. For statistical analysis, since the results involve multi-variate comparison, student's t-test is not sufficient. Please consider using Tukey's test for all the comparisons.

Reviewer #3:

Remarks to the Author:

The major claim that the hepatoprotective effects of Cassiae Semen in animal model was established (<https://www.ncbi.nlm.nih.gov/pmc/articles/PMC6374930/>) and elsewhere (others). However, the correlate this effect in respond to the gut microbiota community is interesting.

1. Figure 2(P) and (Q), 3(N) and (O), and 4(A) and (B) is not informative. I would suggest for you to omit, or rather replace or send to Supplementary section.
2. Line 76 – what is the sequencing depth.
3. Line 81 – Does the effects showed at genera level?
4. Line 175-176 - These effects were presumably exerted by gut microbiota, but which microbiota were involved?. This statements is confusing.
5. Line 371-374 please include the animal study registration number.
6. Line 487 - LEfSE was mentioned but I foresee no related outputs was reported related to the analysis.
7. Line 490 - the term Bacterial metagenome is misleading as this only amplicon sequencing

List of Responses

Dear Editors and Reviewers:

Thank you for your letter and for the reviewers' comments concerning our manuscript entitled "Study on hepatoprotective effects of Cassiae Semen based on the gut microbiota" (ID: COMMSBIO-21-0636-T). Those comments are all valuable and very helpful for revising and improving our paper, as well as the important guiding significance to our researches. We have studied comments carefully and have made a correction which we hope meet with approval. Revised portions are highlighted in the manuscript with a different letter color (red) the changes were performed according to reviewers' comments. The main corrections in the paper and the response to the Editors/reviewers' comments are as following:

Reviewers' comments:

Reviewer #1 (Remarks to the Author):

The authors provide results of an interesting study wherein mice subjected to long-term high-fat diet (HFD) were used as a model of non-alcoholic fatty liver disease (NAFLD). Mice were allowed to develop NAFLD, and then administered treatments consisting of the traditional Chinese medicine (TCM) herb Cassiae semen, or specific extracts from the herb. Outcome measures included serum markers of dyslipidemia, serum ALT and AST as markers of hepatic injury, serum LPS and mRNA levels of two tight junction proteins as markers of gut permeability, a handful of cytokines, and 16S rRNA amplicon sequencing data to assess the fecal microbiota. Following description of the phenotype in HFD-fed mice with and without the herb compounds, the authors perform FMT and antibiotic exposure experiments to support a microbiota-dependent effect of Cassiae semen. While the data strongly support a GM-mediated effect, the exact nature of that effect is still purely correlative. In other words, there are no data presented showing a specific effect of one bacterial taxon or metabolite on NAFLD progression. Rather, the data show nicely that there are multiple features of NAFLD that are reversed (to varying degrees by the tested compounds), and present a wide array of potential changes in microbiota composition and function occurring simultaneously during HFD-induced NAFLD. As such, my biggest overall comment is for authors to review and revise the text critically along these lines, being careful to ascribe too much causality to specific taxa or predicted metabolites.

Major

1. Figure 1 is overly busy and includes information that should be reserved for Results. An example of excessive imagery includes the solid orange horizontal lines which

really add nothing. Text is also excessive in spots, and the figure would be much improved if it contained only the specific experimental treatments and assays performed, without outcomes and descriptions of the treatments. Similarly, panels B and C are filled with abbreviations that have not been addressed in the manuscript yet. All of these should be spelled out in the figure legend.

Response: Thank you very much for your kindly suggestions, figure 1 has been revised. The orange lines should be a display problem, and we have solved it. In addition, some over-expressed concluding sentences have been modified, and the related abbreviations have also been annotated in the figure legend.

2. Related to comment 1, all of these compounds need to be defined at their first usage in the Results (lines 57), and described for the reader. The results describe the effects of these abbreviated compound names, and it took me several minutes to deduce that JMZ was actually the plant (referred to as Cassiae semen in the title so this was unintuitive), as well as the identity of the other compounds being tested. The Introduction should clearly state how and why these specific compounds were selected for testing.

Response: We are very sorry for our unclear expression. According to the Reviewer's suggestion, these extracts and compounds have been defined at their first usage in the text. For better understanding, the Cassiae Semen extracts have been abbreviated as CSE through the full text. For how and why these specific compounds were selected for testing have been claimed in the introduction.

3. Figure panels need to be described in order in the text. Here, panel 2L (line 67) is followed by reference to panel 2P (line 81) and 2Q (line 84). Figure 2M is then presented in line 90.

Response: Thank you for your comment. According to your suggestion, the order of these figure panels has been changed both in the figure and the text.

4. The FMT data are somewhat problematic, and I'm not sure they convincingly show a specific GM-dependent effect. While the FMT recipient receiving material from control mice appear to be protected in terms of many outcomes, the fact that mice receiving material from 'model' mice are also protected is not adequately addressed. Figures 3N and O show that those mice (group FMTM) are protected from the HFD-induced dysbiosis, yet they should be receiving material similar to their existing HFD-induced microbiome, so why do these mice look like control mice? The authors focus on the beneficial effects of the compounds being tested and seem to ignore that transfer of the model (i.e., dysbiotic) microbiota was also protective.

Response: It was speculated that the reason was that, although the abundance of bacteria in the fecal microbial composition of the model group mice is low, they still exist. These low beneficial bacterial levels may act as "primers" and exist in HFD-fed mice. Such

beneficial bacteria generate synergistic effects and affect NAFLD processes via complex mechanisms. Our data suggested the colonization of gut microbiota in mice after FMT was highly complex, therefore further investigation is necessary to understand this behavior.

And figures 3N and O cannot fully reflect the change level of gut microbiota in mice receiving HFD fed mice fecal microbial transplantation, so these two figures have been transferred to the supplementary materials according to the opinions of reviewer 2. Supplementary figure 2A (as shown below) shows the change of gut microbiota on genus level in different FMT groups. Compared with other FMT groups, the microbiota in mice those received fecal microbiological transplantation from HFD-fed mice showed a slightly different trend. For example, *Cupiravidus*, *Bilophila*, and *Roseburia* only increased in FMTH (means those mice who received FMT from HFD-fed mice) group and was no significant change in other groups. Moreover, *Dehalobacterium*, *Odoribater*, *Oscillospira*, and *Roseburia* increased in both FMTC (means those mice who received FMT from control group mice) and FMTCSE (means those mice who received FMT from CSE group mice) groups, while there was no significant change in FMTH groups.

5. The entire antibiotics portion of the Results is frustrating to read, as it's impossible to discern what the groups are without referring to the Methods. In fact, the Results don't even directly state that different types of antibiotics with different spectra were tested. In contrast, there is material in the Methods that is more appropriate in the Introduction or Results (e.g., lines 433-436). Methods should be limited to descriptions of what was done, while descriptions of the rationale and interpretation of tests and experiments belong in the Introduction or Results. The authors are advised to revise the text from the perspective of an average reader that often reads the abstract, and then the introduction and results for further details. Without reading the Methods, can a reader understand what is being done (and why) in each section of the Results?

Response: Thanks a lot for your comment. According to the reviewer's comment, the methods and results of the antibiotic experiment have been corrected accordingly.

6. The portion of the Results speculating on the causal relationships between specific bacteria and NAFLD should be removed or, at a minimum, moved to the Discussion.

While the authors make a good case for the microbiota being at least partially involved in the effects of Cassiae semen, the correlations shown in Figure 5B are just associations, in a largely dichotomous system (i.e., ‘model’ vs all other groups in Fig 5A)

Response: Thanks a lot for your comment. According to the reviewer’s comment, the statements about the causal relationships between specific bacteria and NAFLD have been revised or moved to the Discussion. Fig 5 has been modified as well.

7. Figure 6 is kind of a self-fulfilling prophecy in that you’re showing the correlations between changes in composition and predicted pathways (made using those same changes in composition). Isn’t it to be expected that these outcomes would agree very well? Also, the final statement of the Results is not specifically supported by any data shown here. No metabolites were actually measured and metabolic pathways are inferred using PICRUSt software which is not necessarily the most accurate tool. The same comments apply to the sentence on lines 327-8. These data do not make the predictions any more credible, and the portion of the Discussion describing these correlations over-states the implication of these findings.

Response: Thanks a lot for your very considerate input. The content of microbial community functions prediction has been reanalyzed, including the KO information predicted by the PICRUSt algorithm (annotated by KEGG) and KEGG pathway analysis. The results showed that some results of metabolic function prediction were consistent with our experimental results, such as the increase of LPS biosynthesis pathway in the HFD group was similar to the increase of serum LPS and the Proteobacteria abundance in the HFD group (LPS is the main metabolite of Gram-negative bacteria). More information was shown in the manuscript.

8. Line 306 over-interprets the data, as the data do not definitively show that the hepatoprotective effects of Cassiae semen were due to “improving gut microbiota structures”. While the antibiotic experiments suggest that the microbiota is necessary for the full benefits of Cassiae semen, it’s no conclusive that the fully restored structure is required. The same can be said of the sentence at the start of line 325. It cannot be stated definitively that these genera exacerbated NAFLD, based on these data. Correlations do not necessitate causation.

Response: Thank you very much. These have been revised according to the comment.

Minor

1. On line 32, the Introduction states that Cassiae semen (CS) is used “to relieve constipation, hypolipidemia...” Should this be “hyperlipidemia”, as CS reduces fat accumulation?

Response: Thank you for your reminder, “hypolipidemia” has been revised to “improve hyperlipidemia”.

2. NAFLD should be defined at its first usage in the abstract and main text (line 34).

Response: Thank you very much. It has been revised according to the opinions.

3. The word “flora” (line 42) is somewhat archaic. Consider changing “flora disorder” (same line, also on lines 140, 142, and in Figure 1) to “dysbiosis”.

Response: Thanks a lot for your comment. The full text has been revised according to the comment.

4. For the sake of clarity, I recommend changing “model” to “HFD” throughout (lines 77, 79, 82, etc.) and figures. The word “model” is too vague.

Response: Thank you very much. The full text has been revised according to the comment.

5. Recommend changing the word “executed” in Figure 1 (and anywhere else it appears) to “sacrificed” or “euthanized”.

Response: Thank you very much. It has been revised according to the comment.

6. Suggest adding to line 56 so as to read, “...the effects of Cassiae semen and several of its bioactive compounds on hepatic...”

Response: Thank you for your comment. It has been revised in the text.

7. At first usage, please define host compounds being tested (line 58).

Response: Thank you very much. The full text has been revised according to the comment.

8. What is meant by ‘monomers’ (line 61)? Presumably this refers to RG and AO, but this should be defined for readers (unless I missed it).

Response: Thank you very much. It has been revised in the text.

9. Remove “And” as first word in sentences beginning on line 64 and line 145. In general, sentences should never begin with a conjunction (e.g., and, or, but).

Response: Thank you very much. It has been revised in the text.

10. Correct “occluding” (line 93).

Response: Thank you very much. It has been revised in the text.

11. Authors are advised to overstate the “successful establishment of our antibiotic-induced flora disorder animal model” (lines 140-141, 164). The fact that antibiotics alter the GM isn’t necessarily model validation – what is being modeled? Changes in human GM exposed to same antibiotics?

Response: Thank you very much for your comments. We have made corresponding modifications.

12. Figure 4 is confusing in its use of NA, MA, P, M, etc. for group names in panels A to C, and then switches to different names in the remaining panels. Again, after searching through the Methods, I realized that these are the same groups, just at different time-points. Use a uniform set of group names and specify the differences in the figure legend regarding timing of sample collection.

Response: We are very sorry for our unclear expression. According to reviewer’s

suggestion, the Figure 4 has been modified, and the differences were specified in the figure legend.

13. Authors are advised to remove the word “pharmacodynamics” from line 161 (and elsewhere) as there are no pharmacodynamics data presented here.

Response: Thank you very much. It has been revised in the text.

14. Posing a question to the reader (line 176) is overly colloquial.

Response: We are very sorry for our unclear expression. Considering the Reviewer’s suggestion, we have revised the statement of this part.

15. What is “metastats analysis” (line 177)?

Response: Metastats analysis is a statistical method commonly used in microbiomes, combining nonparametric multiple testing and p-value correction to find species with significant differences between the two groups. (White et al., 2009, doi: 10.1371/journal.pcbi.1000352)

16. Sentence beginning on line 251 belongs in Discussion. Same with sentences from lines 264 to 267, and ending on line 273.

Response: Thank you very much for your comment. We totally agree with your opinion that in-depth discussion should be included in the discussion section, and these sentences were removed to the Discussion. And we also covered the contents of these in-depth discussions in the discussion section.

17. The authors are advised to cite the primary literature showing these beneficial effects (line 294).

Response: Thank you very much for your comment. The primary literatures were cited in the text.

Reviewer #2 (Remarks to the Author):

General comments

This work mainly studied the effects of the traditional Chinese medicine "Jue Ming Zi"(Cassiae Semen, seeds of *Cassia obtusifolia* L. and *C. tora* L) on the hepatoprotective effects via modulation of gut microbiota. According to the authors, Cassiae Semen extract has positive influences on the gut-liver axis, which could produce beneficial metabolites and reduce harmful metabolites. The authors used 16s rDNA sequencing for microbiota compositional analysis and also fecal microbiota transplantation to check whether FMT could transfer the pharmacological effect of Cassiae Semen to HFD-fed mice. The authors made great efforts on the topic and many data were generated. However, the reviewers find that the whole story lacks of integration and the

results part is not thoroughly discussed. In addition, the discussion part lacks of comprehensive analysis. As for the statistical methods used in the study, it is not sufficient. Language issue is also a major problem. Please consider using professional language editing service or ask native English speakers to improve the manuscript. In sum, the reviewer suggests major revision before the manuscript could be considered for publication on Communications Biology.

Minor issues

1. Line 6. No verb in the sentence of "Cassiae Semen, seeds of Cassia obtusifolia L. and C. tora L, so-called "Jue-Ming-Zi" in Chinese." Please revise the sentence into "Cassiae Semen, seeds of Cassia obtusifolia L. and C. tora L, is also known as "Jue-Ming-Zi" in Chinese."

Response: Thank you very much. It has been revised in the text.

2. Line 7. "Cassiae semen relieves constipation and exerts hepatoprotective effects" Please pay attention to third-person singular.

Response: Thank you very much for your reminder. It has been revised in the text.

3. Line 10. "We examined the relationships... and mechanisms..."

Response: Thank you very much for your carefully review. It has been revised in the text.

4. Line 12 Our data... inflammation, hence having a good protective effect on"

Response: Thank you very much. It has been revised in the text.

5. When using the abbreviation NAFLD for the first time, please give the full name of the abbreviation. For example, "Non-alcoholic fatty liver disease (NAFLD)"

Response: Thank you very much. It has been revised in the text.

6. Line 21 "Consumption of ...with fat accumulating, increasing the risk of...". It is not correct to use "increased the risk"

Response: Thank you very much for your carefully review. It has been revised in the text.

7. Line 22-24: verb is missing in the sentence. Please check the manuscript and revise this type of grammar mistake. Please also consider using professional language editing service for the manuscript.

Response: We are so sorry for our grammar mistake. And the manuscript has been sent to professional language editing service company for the spell and grammar checking.

8. Line 30-31: please delete "in China" due to the repetition.

Response: Thank you very much for your carefully review. It has been revised in the text.

9. Line 44: The experimental setup is shown in Fig. 1. Please remove the brackets ().

Response: Thank you very much for your carefully review. It has been revised in the

text.

10. There are many abbreviations used in the figure 1. Could the authors add the full term-abbreviation pair in the figure legend. For example, HFD:high fat diet, NAFLD: non-alcoholic fatty liver disease...

Response: Thank you very much. The full term-abbreviation pair have been added in the figure legend.

11.Line 56. "C57BL/6 mice were fed with HFD for..."

Response: Thank you very much for your carefully review. It has been revised in the text.

12. What is JMZ, TA, RG, and AO? When using the abbreviations for the first time, please give the full terms.

Response: Thank you very much. These have been revised in the text.

13. Line 78. After administration of "what"?

Response: We are so sorry for our unclear expression. It should be "After Cassiae Semen extracts or monomers administration...", and it has been revised in the text.

14. Line 124: please make sure that the tenses in the same sentence are the same. "Our data suggested the colonization of gut microbiota in mice after FMT was highly complex. Thus, more research should be required to prove our hypothesis".

Response: Thank you for your carefully review. These have been revised in the text.

15. For statistical analysis, since the results involve multi-variate comparison, student's t-test is not sufficient. Please consider using Tukey's test for all the comparisons.

Response: Thank you for your suggestion, and the Tukey multiple comparison test was used for multi-variate comparison, the corresponding content has been modified in the text.

Reviewer #3 (Remarks to the Author):

The major claim that the hepatoprotective effects of Cassiae Semen in animal model was established (<https://www.ncbi.nlm.nih.gov/pmc/articles/PMC6374930/>) and elsewhere (others). However, the correlate this effect in respond to the gut microbiota community is interesting.

1. Figure 2(P) and (Q), 3(N) and (O), and 4(A) and (B) is not informative. I would suggest for you to omit, or rather replace or send to Supplementary section.

Response: Thank you for your suggestion. According to your comments, these figures have been sent to Supplementary section.

2. Line 76 – what is the sequencing depth.

Response: Sequencing depth is a term in genomic analyses, which means the average number of times that a particular nucleotide is represented in a collection of random

raw sequences. Please refer to this article (<https://www.nature.com/articles/nrg3642>) for details.

3. Line 81 – Does the effects showed at genera level?

Response: In this part, we mainly discussed the main changes in the phylum level of gut microbiota, and the effects at genera level were discussed in the later part “Key gut microbiota altered by Cassiae Semen during NAFLD treatment”.

4. Line 175-176 - These effects were presumably exerted by gut microbiota, but which microbiota were involved? This statements is confusing.

Response: We are very sorry for our unclear expression. Considering the Reviewer’s suggestion, we have revised the statement of this part.

5. Line 371-374 please include the animal study registration number.

Response: Thank you for your reminder. The animal study registration number was added in the text. The project identification code was 20192005.

6. Line 487 - LEfSE was mentioned but I foresee no related outputs was reported related to the analysis.

Response: We used LEfSe analysis in the previous version of the manuscript and then it was omitted. Thank you very much for your comments. This part has been revised.

7. Line 490 - the term Bacterial metagenome is misleading as this only amplicon sequencing

Response: Thank you very much for your comments. We have made relevant modifications.

REVIEWERS' COMMENTS:

Reviewer #1 (Remarks to the Author):

Thank you for addressing my comments and concerns.

Reviewer #2 (Remarks to the Author):

The authors have satisfactorily answered all my questions and concerns. I would like to endorse the publication of this manuscript.

List of Responses

REVIEWERS' COMMENTS:

Reviewer #1 (Remarks to the Author):

Thank you for addressing my comments and concerns.

Response: Thank you for your recognition of our work, and thanks again for your valuable comments.

Reviewer #2 (Remarks to the Author):

The authors have satisfactorily answered all my questions and concerns. I would like to endorse the publication of this manuscript.

Response: Thanks again for your very helpful comments.